

# DNMT family induces down-regulation of NDRG1 *via* DNA methylation and clinicopathological significance in gastric cancer

Xiaojing Chang[1,*], Jinguo Ma[2,*], Xiaoying Xue[1], Guohui Wang[1], Tianfang Yan[3], Linlin Su[1], Xuetao Han[1], Huandi Zhou[4] and Liubing Hou[4]

[1] Department of Radiotherapy, The Second Hospital of Hebei Medical University, Shijiazhuang, China
[2] Department of Internal-Oncology, Hulun Buir People's Hospital, Hulun Buir Medical School, Nationalities University of Inner Mongolia, Hulun Buir, China
[3] Department of Neurological Diagnosis and Restoration, Osaka University Graduate School of Medicine, Osaka, Japan
[4] Department of Central Laboratory, The Second Hospital of Hebei Medical University, Shijiazhuang, China
* These authors contributed equally to this work.

## ABSTRACT

**Background:** Aberrant DNA methylation of tumor suppressor genes is a common event in the development and progression of gastric cancer (GC). Our previous study showed NDRG1, which could suppress cell invasion and migration, was frequently down-regulated by DNA methylation of its promoter in GC.

**Purpose and Methods:** To analyze the relationship between the expression and DNA methylation of NDRG1 and DNA methyltransferase (DNMT) family. We performed a comprehensive comparison analysis using 407 patients including sequencing analysis data of GC from TCGA.

**Results:** NDRG1 was down-regulated in GC, and was negatively correlative to DNMT1 ($r = -0.11$, $p = 0.03$), DNMT3A ($r = -0.10$, $p = 0.01$), DNMT3B ($r = -0.01$, $p = 0.88$), respectively, whereas the DNA methylation of NDRG1 was positively correlative to DNMT family (DNMT1 $r = 0.20$, $p < 0.01$; DNMT3A $r = 0.26$, $p < 0.001$; DNMT3B $r = 0.03$, $p = 0.57$, respectively). NDRG1 expression was significantly inverse correlated with invasion depth ($p = 0.023$), but DNMT1 was significantly positive correlated with invasion depth ($p = 0.049$). DNMT3B was significantly correlated with the degree of tumor cell differentiation ($p = 0.030$). However, there was no association between the expression of DNMT3A and clinicopathological features. The KM plotter showed that NDRG1 (HR = 0.95, 95% CI [0.8–1.12], $p = 0.53$) and DNMT1 (HR = 1.04, 95% CI [0.88–1.23], $p = 0.67$) had no association with prognosis of GC patients, while, DNMT3A ($p = 0.0064$) and DNMT3B ($p = 0.00025$) displayed significantly association. But the overall survival of high expression of NDRG1 tended to be prolonged.

Corresponding author
Xiaoying Xue, xxy0636@163.com

**Conclusion:** These data suggest that down-regulation of NDRG1expression in GC may be due to its promoter DNA methylation *via* DNMT family. The demethylating agent maybe a potential target drug for GC patients.

## INTRODUCTION

Gastric cancer (GC) is one of the most common digestive malignancies worldwide with the fourth most common cause of cancer-related death (*Sung et al., 2021*; *Ferlay et al., 2019*). The poor prognosis and high recurrence rate are mainly due to lymph node metastasis at its early stage.

To date, a number of laboratories have shown that the occurrence of GC is usually caused by oncogenes activation and tumor suppressor genes (TSGs) inactivation. TSGs could suppress tumor cells migration and invasion, studies have shown that the inactivation of most TSGs are related to DNA methylation of CpG, while the silencing of these genes possibly contributes to the development and progression of tumors (*Song et al., 2012*; *Celarain & Tomas-Roig, 2020*; *Hackett & Surani, 2013*). There is increasing evidence that aberrant DNA promoter methylation of TSGs is a major event in the development and progression of GC (*Qu, Dang & Hou, 2013*). Our previous studies showed NDRG1 (N-myc downstream-regulated gene 1), a differentiation-related and tumor suppressor gene, which could suppress cancer cells invasion and migration, was frequently down-regulated by DNA methylation of its promoter in GC, while there were not any mutations in 16-exon sequences of NDRG1, and also not significant correlation with histone acetylation (*Chang, 2013*, *2014*).

DNA methylation of CpG islands, which is carried out by DNA methyltransferase (DNMT) enzymes, is the most widely and the best well studied epigenetic modification event, and leads to transcriptional gene silencing (*Lyko, 2018*). DNMT enzymes include DNMT1, DNMT2, DNMT3A, DNMT3B and DNMT3L, among which DNMT1 (a maintenance DNMT), DNMT3A and DNMT3B (*de novo* methyltransferases) are the most important (*Baylin & Jones, 2011*). In the current study, we focused on the relationship between NDRG1 and DNMT family (DNMT1, DNMT3A and DNMT3B) and their clinical significance and prognosis in GC by analyzing high-throughput data obtained from TCGA.

## MATERIALS AND METHODS

### The datasets of GC

The data that we investigated the mRNA expression of NDRG1 and DNMT family members (DNMTs), and the DNA methylation of NDRG1 in GC were obtained from the data portal of TCGA (https://portal.gdc.cancer.gov/), which is a well-known cancer research project that collects and analyzes high-quality tumor samples and makes the related data available to researchers. We obtained the GC data set (March 1st, 2018

**Table 1 Primers of NDRG1 and DNMT family.**

| Gene | Primer |
| --- | --- |
| NDRG1 | 5′-AGTTGGAGTAAGTGTTTGTGAGT-3′ (sense) |
|  | 5′-AAACAAAAACAAAAATAACACCACT-3′ (antisense) |
| DNMT1 | 5′-AGGCGGCTCAAAGATTTGGAA-3′ (sense) |
|  | 5′-GCAGAAATTCGTGCAAGAGATTC-3′ (antisense) |
| DNMT3A | 5′-CCGATGCTGGGGACAAGAAT-3′ (sense) |
|  | 5′-CCCGTCATCCACCAAGACAC-3′ (antisense) |
| DNMT3B | 5′-AGGGAAGACTCGATCCTCGTC-3′ (sense) |
|  | 5′-GTGTGTAGCTTAGCAGACTGG-3′ (antisense) |
| GAPDH | 5′-CATGAGAAGTATGACAACAGCCT-3′ (sense) |
|  | 5′-AGTCCTTCCACGATACCAAAGT-3′ (antisense) |

updated), which encompassed 407 gastric tumor samples, and of which 338 cases encompassed the data of DNA methylation of NDRG1. A total of 315 cases contained the complete clinicopathological information. GSE13911 data sets including 69 cases (including 38 GC tissues) of GC patients with microarray expression spectrum and corresponding clinical information were downloaded from the GEO (https://www.ncbi.nlm.nih.gov/geo/). Data from the TCGA and GEO database were free for using in any research study and did not require a separate consent from the participants. The present study was in compliance with the 1964 Declaration of Helsinki and its later amendments or comparable ethical standards.

## Prognostic analysis of NDRG1 and DNMT family *via* Kaplan–Meier (KM) plotter

The prognostic values of overall survival (OS) for NDRG1 and DNMT family (RNAseq data) in GC ($n = 875$) were analyzed by KM plotter (http://kmplot.com/analysis/index.php?p=service&cancer=gastric). The high/low expression of NDRG1 and each DNMT member were determined by the median value. The results were displayed by hazard ratio (HR), 95% confidence intervals (95% CI) and log-rank $p$ value (significant threshold was <0.05).

## Gastric cells culture and real-time PCR

Human gastric cancer cell lines, SGC7901 and MKN45, and one immortalized normal gastric cell line, GES1 were cultured as described in our previous study (*Chang, 2013*).

Total RNA was isolated and reversely transcribed into cDNA as described previously (*Chang, 2013*). The polymerase chain reaction (PCR) was performed in a volume of 20 ul using MonAmp™ ChemoHS qPCR Mix (Monad Biotech Co., Ltd, China) with an initial denaturing at 95 °C for 10 min, followed by 40 cycles of denaturing at 95 °C for 10 s, annealing for 10 s at 60 °C, extension for 30 s at 72 °C. The primers were shown in Table 1. Relative gene expression to an internal GAPDH control and, hence, fold changes were calculated using the equation $2^{-\Delta\Delta Ct}$ method.

## Protein isolation and western blot

Total cellular protein was extracted and western bolt was performed and anti-NDRG1 antibody was used as previous description (*Chang, 2013*). Protein bands were scanned and quantified using densitometric software (Bio-Rad, Irvine, CA, USA). DNMT1, DNMT3A and DNMT3B antibodies at a dilution of 1:500 were purchased from Bioss, Beijing, China.

## Statistics

All statistical analyses were conducted using SPSS 24.0 (Chicago, IL, USA) and R 3.6.1 (https://www.r-project.org/). R language and GraphPad Prism 7 (San Diego, CA, USA) were performed to draw plots. $X^2$ test and Fisher's exact test were used to generate *P* values for **clinicopathological parameters**, Values with $p < 0.05$ were considered as statistically significant.

# RESULTS

## The expression of NDRG1 and DNMT family in GC

Total of 407 GC cases (375 were GC tissues and 32 were nonmalignant gastric tissues) which included sequencing analysis data were downloaded from TCGA, then we firstly performed a expression analysis about the NDRG1 and DNMTs. As shown in Figs. 1A–1D, NDRG1 mRNA expression was down-regulated, but all of DNMT1, DNMT3A and DNMT3B were up-regulated in GC tissues compared with normal tissues ($p < 0.05$). To confirm these results, we further downloaded sequencing analysis data of GC from GEO which included 69 GC cases (38 were GC tissues and 31 were nonmalignant gastric tissues), results were similar to that of TCGA, but the expression of NDRG1 and DNMT3A showed no statistical significance due to the small sample size (Figs. 2A–2D).

We also assessed the mRNA and protein levels of NDRG1 and DNMTs in one immortalized normal gastric cell line, GES1 and two gastric cancer cell lines, SGC7901 and MKN45 by real-time PCR and western blot. Results showed DNMT1, DNMT3A and DNMT3B mRNA expression were up-regulated in SGC7901 (1.63-fold, 2.34-fold, 2.14-fold, respectively) and MKN45 (1.77-fold, 3.62-fold, 1.87-fold, respectively) cells compared with GES1 (one fold as the control). All of DNMTs mRNA levels were negatively correlative to the expression of NDRG1 which was shown down-regulated in GC cells (including SGC7901 and MKN45 cell lines) in our previous study (*Chang, 2013*) ($p < 0.05$, Fig. 3A). Western blot analysis confirmed our data on NDRG1 and DNMTs mRNA expression (Fig. 3B). This was in accordance to the results of the relationship between NDRG1 and DNMT family mRNA expression from TCGA.

## NDRG1 expression was negatively correlative to DNMT family, but its promoter methylation was positively correlative to the expression DNMT family in GC

We further performed a comprehensive comparison analysis about the mRNA expression of NDRG1 and DNMTs. As shown in Figs. 4A–4C, NDRG1 mRNA expression was negatively correlative to DNMT1 ($r = -0.11$, $p = 0.03$), DNMT3A ($r = -0.10$, $p = 0.01$),

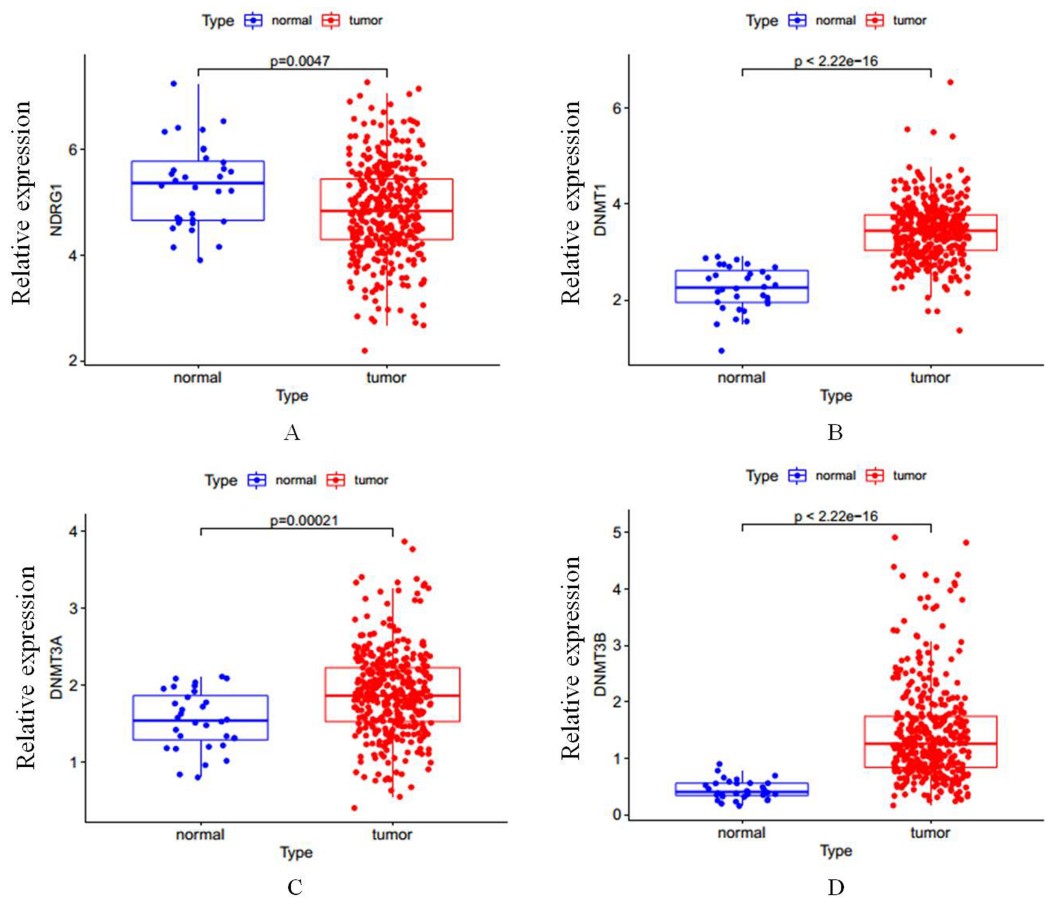

**Figure 1 (A–D) The mRNA expression of NDRG, DNMT1, DNMT3A and DNMT3B in GC.**

DNMT3B (r = −0.01, $p$ = 0.89). Although the R value of comparison analysis was low, it still had statistical significance. NDRG1 was significantly associated with DNMT1 ($p$ = 0.03) and DNMT3A ($p$ = 0.01), and DNMT3A showed the strongest association. However, there was no significant association between NDRG1 and DNMT3B ($p$= 0.89).

Among all of 407 GC cases, 338 cases encompassed the data of DNA methylation of NDRG1. Results showed DNA methylation of NDRG1 gene promoter was positively correlative to the expression DNMT family, and was significantly associated with DNMT1 (r = 0.20, $p$ = 0.002, Fig. 4D) and DNMT3A (r = 0.26, $p$ < 0.001, Fig. 4E). However, no statistical significance was found between NDRG1 and DNMT3B (r = 0.03, $p$ = 0.57, Fig. 4F). Although the R value of comparison analysis was not high, it still had statistical significance. It suggested that the down-regulation of NDRG1 in GC may be due to DNA methylation of its promoter *via* DNMT family (DNMT1 and/or DNMT3A).

## The association of NDRG1 and DNMT family with the clinicopathological parameters and prognosis of GC

The association of NDRG1 and DNMT family with the clinicopathological parameters of GC patients were shown in Table 2. A total of 315 cases which contained the complete

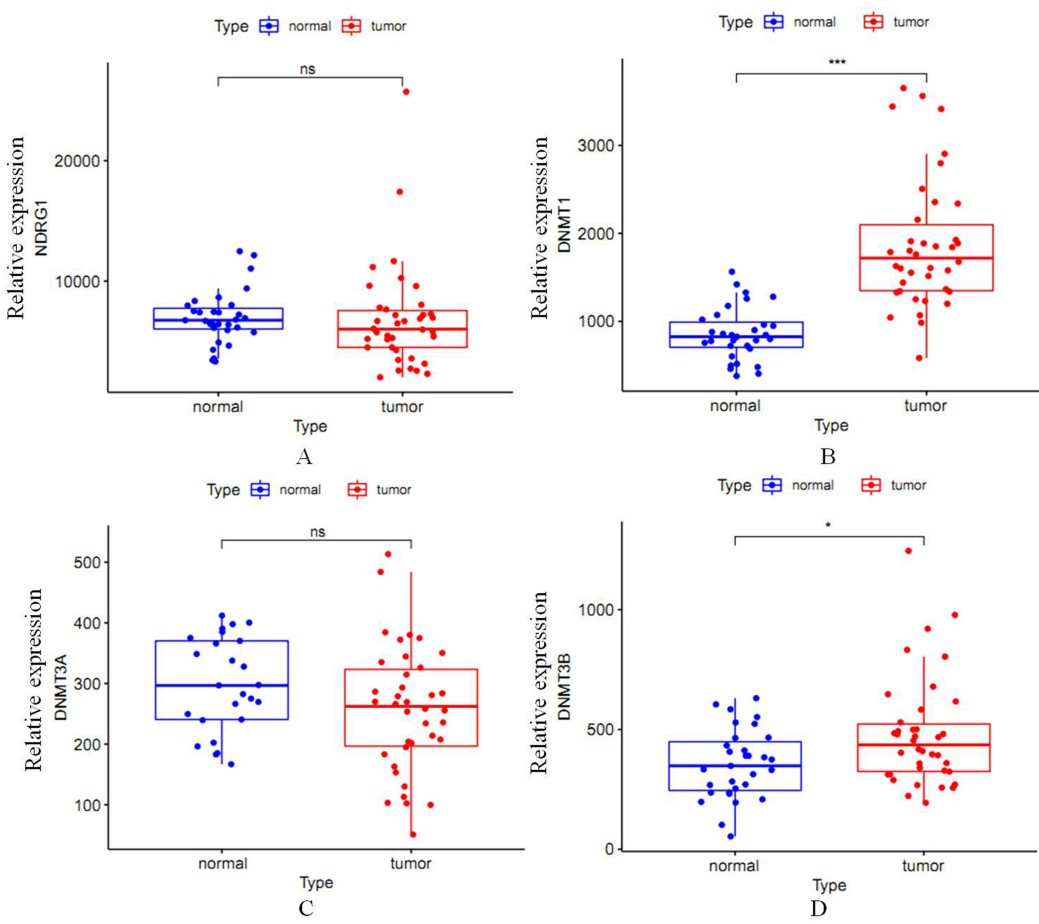

**Figure 2** **(A–D) The mRNA expression of NDRG1, DNMT1, DNMT3A and DNMT3B in GC from the data of GEO.** $*p < 0.05$; $***p < 0.001$; ns, no statistical significance.

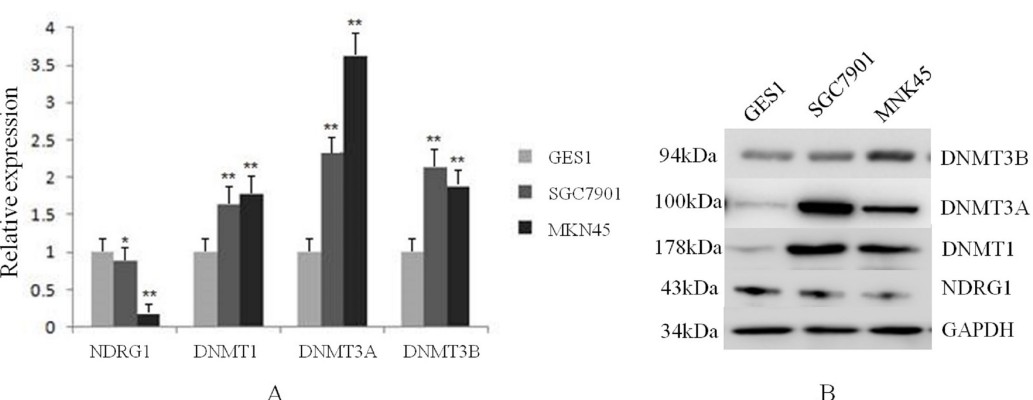

**Figure 3** **The expression levels of NDRG1 and DNMTs in GES1 and two gastric cancer cell lines, SGC7901 and MKN45.** (A) The mRNA level of NDRG1 and DNMTs; (B) The protein level of NDRG1 and DNMTs ($*p < 0.05$, $**p < 0.01$).

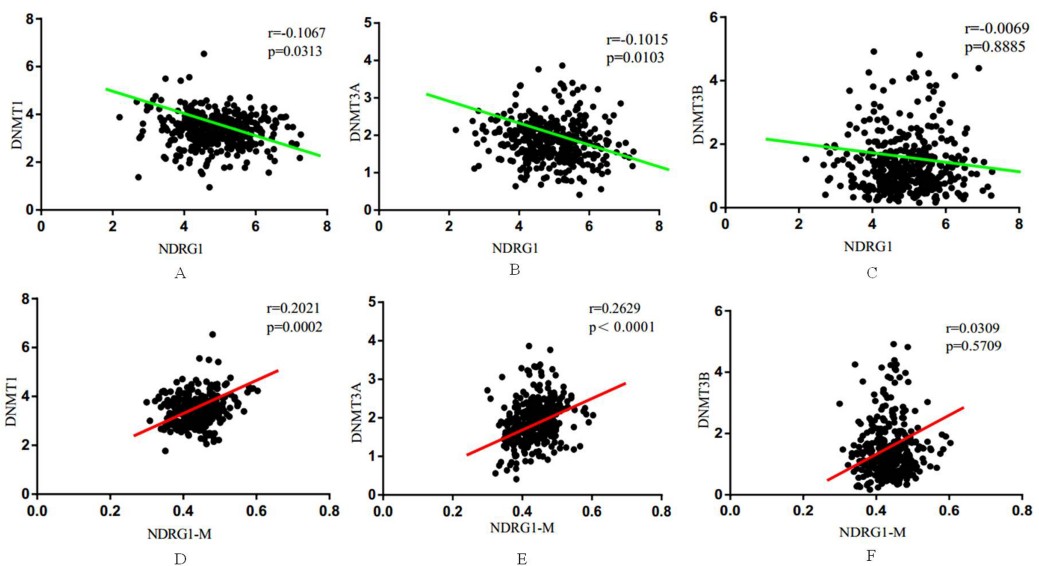

**Figure 4 Association between NDRG1 and DNMTs.** (A–C) NDRG1 mRNA expression was negatively correlative to DNMT1, DNMT3A and DNMT3B. (D–F) DNA methylation of NDRG1 gene promoter was positively correlative to the expression DNMT family.

clinicopathological and survival information were analyzed. The high and low expression of NDRG1 and DNMTs were based on the median value of their mRNA levels. We found NDRG1 expression was significantly inverse correlated with invasion depth ($p = 0.023$), whereas DNMT1 was statistically significantly positive correlated with invasion depth ($p = 0.049$), DNMT3B was significantly positive correlated with the degree of tumor cell differentiation ($p = 0.030$). There was no significant association between DNMT3A and the clinicopathological parameters of the GC patients (Table 2).

The prognostic values of NDRG1 and DNMTs in GC were assessed *via* the KM plotter. Results showed that NDRG1 (HR = 0.95, 95% CI [0.8–1.12], $p = 0.53$) and DNMT1 (HR = 1.04, 95% CI [0.88–1.23], $p = 0.67$) showed no significantly association with OS of GC patients. Whereas, DNMT3A (HR = 1.35, 95% CI [1.09–1.67], $p = 0.0064$) and DNMT3B (HR = 1.37, 95% CI [1.16–1.63], $p = 0.00025$) were significantly correlated with prognosis of GC patients (Figs. 5A–5D). But for median OS, it seemed that high expression of NDRG1 displayed good outcome (29.5 vs 27.8 months) compared with its low expression, while high expression of DNMT1, DNMT3A and DNMT3B showed poor prognosis.

## DISCUSSION

NDRG1, a differentiation-related gene, which belongs to NDRG family, is down-regulated in many tumors, and proposed as a metastasis suppressor gene (*Wang et al., 2020*; *Chen et al., 2018*; *Park et al., 2019*; *Menezes, Kovacevic & Richardson, 2019*). Our previous studies confirmed the anti-cancer effect of NDRG1 in GC (*Chang, 2013*; *Chang, 2014*; *Chang et al., 2016*). Gene network analysis shows that the 5 'end of the NDRG1 gene contains a large CpG island. It was reported that aberrant DNA methylation of CpG could

**Table 2 Clinicopathological parameters of NDRG1 and DNMTsmRNA expression from TCGA cohort.**

| Variable | Patients (n) | NDRG1 Low (High) | p | DNMT1 Low (High) | p | DNMT3A Low (High) | p | DNMT3B Low (High) | p |
|---|---|---|---|---|---|---|---|---|---|
| Gender | | | | | | | | | |
| Female | 120 | 61 (59) | 0.908 | 56 (64) | 0.355 | 56 (64) | 0.355 | 61 (59) | 0.817 |
| Male | 195 | 97 (98) | | 102 (93) | | 102 (93) | | 96 (99) | |
| Age | | | | | | | | | |
| ⊠60 | 100 | 52 (48) | 0.717 | 47 (53) | 0.469 | 50 (50) | 1.000 | 49 (51) | 0.904 |
| ≥60 | 215 | 106 (109) | | 111 (104) | | 108 (107) | | 108 (107) | |
| Stage | | | | | | | | | |
| I | 42 | 18 (24) | 0.297 | 23 (19) | 0.881 | 26 (16) | 0.190 | 20 (22) | 0.814 |
| II | 101 | 47 (54) | | 51 (50) | | 51 (50) | | 49 (52) | |
| III | 139 | 78 (61) | | 69 (70) | | 62 (77) | | 69 (70) | |
| IV | 33 | 15 (18) | | 15 (18) | | 19 (14) | | 19 (14) | |
| Invasion depth | | | | | | | | | |
| T1 | 15 | 9 (6) | 0.023* | 12 (3) | 0.049* | 11 (4) | 0.090 | 9 (6) | 0.216 |
| T2 | 63 | 21 (42) | | 33 (30) | | 37 (26) | | 28 (35) | |
| T3 | 151 | 84 (67) | | 77 (74) | | 69 (82) | | 70 (81) | |
| T4 | 66 | 44 (42) | | 36 (50) | | 41 (45) | | 50 (36) | |
| Lymph node metastasis (N) | | | | | | | | | |
| N0 | 99 | 46 (53) | 0.348 | 49 (50) | 0.828 | 52 (47) | 0.864 | 50 (49) | 0.206 |
| N1 | 83 | 48 (35) | | 40 (43) | | 43 (40) | | 44 (39) | |
| N2 | 69 | 31 (38) | | 38 (31) | | 33 (36) | | 27 (42) | |
| N3 | 64 | 33 (31) | | 31 (33) | | 30 (34) | | 36 (28) | |
| Distant metastasis | | | | | | | | | |
| No | 295 | 149 (146) | 0.652 | 149 (146) | 0.652 | 145 (150) | 0.248 | 144 (151) | 0.174 |
| Yes | 20 | 9 (11) | | 9 (11) | | 13 (7) | | 13 (7) | |
| Differentiation | | | | | | | | | |
| G1 | 7 | 3 (4) | 0.293 | 4 (3) | 0.906 | 3 (4) | 0.758 | 3 (4) | 0.030* |
| G2 | 108 | 48 (60) | | 55 (53) | | 57 (51) | | 43 (65) | |
| G3 | 200 | 107 (93) | | 99 (101) | | 98 (102) | | 111 (89) | |

Note:
* $p < 0.05$.

lead to transcriptional gene silencing (*Lyko, 2018*). Some studies have confirmed the downregulation of NDRG1 caused by DNA methylation of CpG islands of its promoter in breast cancer, prostate cancer cells and pancreatic cancer (*Han, 2013*; *Li, 2015*; *Angst, 2010*). In GC, the down-regulation of NDRG1 was also found to be regulated by DNA methylation of its promoter in our previous study (*Chang, 2013*). DNA methylation, which is carried out by DNA methyltransferase enzymes, plays important roles in the regulation of gene expression and the development and progression of tumors. The DNMT family includes DNMT1, DNMT2, DNMT3A, DNMT3B, and DNMT3L, among which DNMT1, DNMT3A and DNMT3B are the most important members (*Baylin & Jones, 2011*). To date, not any studies reported the relationship of NDRG1 and DNMT family in GC.

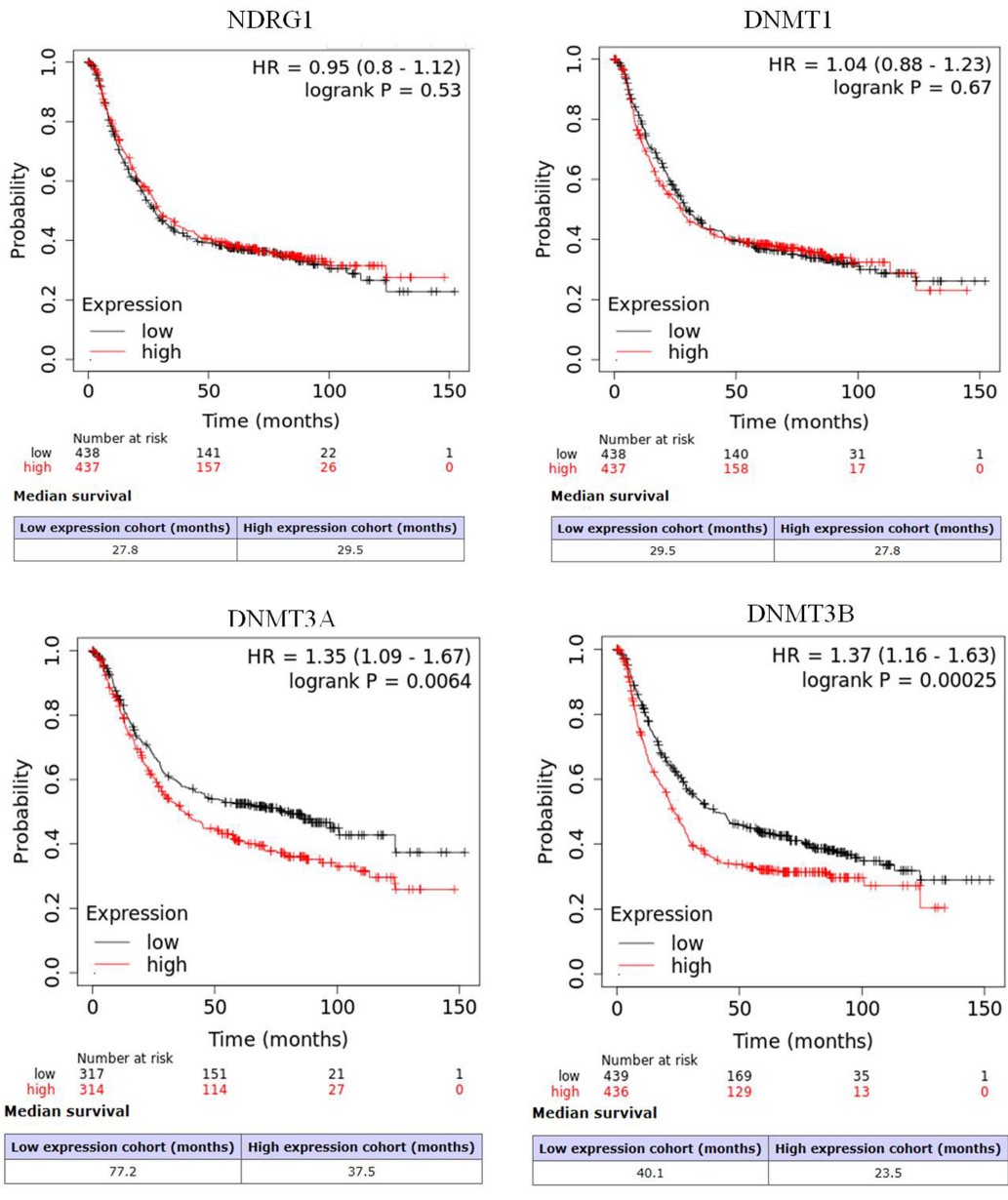

**Figure 5 Survival curves of NDRG1 and DNMT family in GC *via* the Kaplan–Meier (KM) plotter.**
Red: high expression; black: low expression; HR: hazard ratio.

In the current study, a total of 407 GC cases which included sequencing analysis data of NDRG1 mRNA were analyzed. We found that all of DNMT1, DNMT3A and DNMT3B were up-regulated in GC, while, NDRG1 mRNA level was down-regulated in GC tissues. It is similar to our previous study (*Chang, 2013*). Data from GEO and results of PCR and western blot analysis from GC cell lines confirmed these data from TCGA. Results of comparison analysis showed that NDRG1 expression was negatively correlative to DNMT1, DNMT3A and DNMT3B, and DNMT3A showed the strongest association. Although the R value of comparison analysis was not high, it still had statistical
significance. But no significant association was found between NDRG1 and DNMT3B. This is the first time for analyzing the relationship between NDRG1 and DNMT family in GC. NDRG2, another member of NDRG family which includes four members NDRG1-4, was reported to be significantly decreased in human cancers, and in GC, *H. pylori* silenced Ndrg2 by activating the NF-κB pathway and up-regulating DNMT3b, promoting GC progression (*Ling, 2015*). Our current data were similar to those of NDRG2. Noteworthy, our data showed that aberrant DNA methylation of NDRG1 may be mainly due to the regulation of DNMT1 and DNMT3A.

We further investigated the association of DNA methylation of NDRG1 and DNMTs. Total 338 of all 407 GC cases encompassed the data of DNA methylation of NDRG1. Conversely, DNA methylation of NDRG1 was positively correlative to the expression of DNMTs, and was significantly associated with DNMT1 and DNMT3A.These results once again confirmed the relationship of NDRG1 mRNA expression and DNMTs in GC. Our previous study found that down-regulation of NDRG1 was regulated by DNA methylation of its promoter. DNA methylation of CpG islands leads to transcriptional gene silencing which is carried out by DNMTs. Thus, our current data suggests that down-regulation of NDRG1 in GC may be due to DNA methylation of NDRG1 gene promoter *via* DNMT family, whereas NDRG1 expression was significantly inverse correlated with invasion depth, and DNMT1 was positively correlated with invasion depth. DNMT3B was significantly positive correlated with the degree of tumor cell differentiation. However, KM plotter showed no association between NDRG1and DNMT1 with the prognosis of GC patients, while high expression of DNMT3A and DNMT3B were significantly correlated with poor prognosis of GC patients. But for median survival, it seemed that high expression of NDRG1 displayed good outcome and the OS tended to be prolonged. These results were mainly based on the data of bioinformatics, the current data remain suggest that high NDRG1 expression and low DNMTs expression could suppress cell invasion and migration, may improve the rate of recurrence and metastasis. Our previous study had showed GC patients with high expression of NDRG1 had better overall survival rate than those with low NDRG1 expression (*Chang, 2014*). Therefore, large sample data of immunohistochemistry, western blot and RT-PCR of GC tissues are urgently needed.

DNMT inhibitor, including azacitidine, 5-aza-2′-deoxycytidine (5-Aza-Dc, decitabine), guadecitabine, hydralazine, procaine, MG98 and/or zebularine, among which 5-Aza-Dc is the most commonly used, could cause demethylating and reactivate the expression of TSGs, then suppress the metastasis of tumor cells (*Cheishvili, 2015*; *Zhou, 2018*; *Katarzyna & Lucyna, 2019*). NDRG1 expression was found to be increased after treatment with 5-Aza-Dc in breast cancer, prostate cancer and pancreatic cancer (*Han, 2013*; *Li, 2015*; *Angst, 2010*). *Lee et al. (2018)* found that the epidrug 5-Aza could up-regulated NDRG1 expression by reduction of suppressive histone marks, H3K9me3 and H3K27me3 on NDRG1 promoter in prostate cancer.

In the current study, the aberrant DNA methylation of NDRG1 was found to maybe significantly associated with DNMT family, especially DNMT1 and DNMT3A. Our previous study showed that the down-regulation of NDRG1 was also found to be
**Table 3 Ongoing clinical trials evaluating decitabine drug in solid tumors.**

| Study number | Phase | Drug | Disease | Status completion date |
|---|---|---|---|---|
| NCT03875287 | I | Decitabine Cedazuridine | Solid tumors | Recruiting August 2021 |
| NCT04049344 | II | Combination of decitabine with oxaliplatin | Relapsed/metastatic renal cell carcinoma | Recruiting December, 2021 |
| NCT03445858 | I | Pembrolizumab combined with Decitabine | Young adult patients with relapsed and refractory solid tumors or lymphoma | Recruiting January 12, 2025 |
| NCT02959164 | Ib | Decitabine and Gemcitabine | Refractory pancreatic adenocarcinoma and advanced soft tissue or bone sarcomas | Active, June, 2021 not recruiting |

regulated by DNA methylation of its promoter and 5-Aza-Dc could up-regulate its expression in GC cells. It suggests that the demethylating agent maybe a potential target drug in GC. Up to now, demethylating treatment has been used for the treatment of myeloid leukemias (*Sato, Issa & Kropf, 2017*; *Dombret & Itzykson, 2017*). *In vitro* studies of solid tumors, demethylating agents combined chemotherapy and/or immunotherapy could enhance the therapeutic effect in cancer cells (*Li et al., 2013*; *Lou et al., 2014*; *Vijayaraghavalu & Labhasetwar, 2018*; *Luo, Sugiura & Balko, 2018*). In a case report, azacitidine was used to treat a 57-year-old woman newly diagnosed with MDS during palliative chemotherapy for metastatic breast cancer, azacitidine showed promising effects for MDS, and also stabilized the patient's lung and lymph node metastases without any major toxicity (*Baek et al., 2019*). An increasing number of clinical trials of demethylating treatment is ongoing, some of which have presented exciting effects (*Linnekamp et al., 2017*; *Perrard et al., 2020*; *Singal et al., 2015*; *Jansen et al., 2019*) (Table 3). A Phase I study showed the security of decitabine treated by hepatic arterial infusion, and a dose level of 20 mg/m$^2$/day on five consecutive days every 4 weeks could be considered for further investigation in combinatorial immunotherapy regimens in patients with Unresectable Liver-predominant Metastases From Colorectal Cancer (*Singal et al., 2015*). In a phase I/II clinical trial, 15 patients with metastatic castration-resistant prostate cancer (mCRPC) were enrolled in phase I which was dose climbing experiment of 5-Aza-Dc, and 7 patients were enrolled in phase II. In phase I, no dose-limiting toxicity was observed, and most patients were well tolerated. The highest level reached was 5-Aza-Dc with 150 mg/m$^2$ daily for 5 days followed by docetaxel with 75 mg/m$^2$ on day 6, which was considered as the recommended phase II dose. In phase II, six patients received the recommended phase II dose during 46 cycles. Two episodes of Grade 3 hematologic and three Grade 3 nonhematologic toxicities were observed, and one patient died from neutropenic sepsis. Subsequently, 5-Aza was reduced to 75 mg/m$^2$, and no treatment-related > Grade 3 toxicities was observed in one patient. In this clinical trial, PSA response was observed in 10 of 19 (52.6 %) patients, and the median duration of response was 20.5 weeks. Kaplane–Meier estimate of median PFS was 4.9 months, and median OS was 19.5 months, which were both favorable (*Singal et al., 2015*). It suggests that the combination of azacitidine and chemotherapy is active in mCRPC patients, and it may be a new treatment in future for tumors.

This study is mainly based on the data of bioinformatics; a large sample data of immunohistochemistry, western blot and RT-PCR of GC tissues are urgently needed. We will further detect and confirm the relationship of NDRG1 and DNMTs in GC tissues and *in vivo* and vitro experiments.

In conclusion, the current data showed that the down-regulated and its aberrant DNA methylation of NDRG1 which could suppress cell invasion and migration, maybe mainly regulated by DNMT family in GC, especially DNMT1 and DNMT3A. Furthermore, the demethylating agent 5-Aza-Dc, a DNMT inhibitor, maybe a potential target drug. Therefore, further clinical studies are warranted to evaluate and confirm the effect of demethylating treatment in GC.

### Funding
This work was supported by the National Natural Science Foundation of Hebei province of China (No. H2018206180). There was no additional external funding received for this study. The funders had no role in study design, data collection and analysis, decision to publish, or preparation of the manuscript.

### Grant Disclosures
The following grant information was disclosed by the authors:
National Natural Science Foundation of Hebei province of China: H2018206180.

### Competing Interests
The authors declare that they have no competing interests.

### Author Contributions
- Xiaojing Chang performed the experiments, prepared figures and/or tables, authored or reviewed drafts of the paper, and approved the final draft.
- Jinguo Ma performed the experiments, prepared figures and/or tables, and approved the final draft.
- Xiaoying Xue conceived and designed the experiments, prepared figures and/or tables, and approved the final draft.
- Guohui Wang performed the experiments, analyzed the data, prepared figures and/or tables, and approved the final draft.
- Tianfang Yan performed the experiments, prepared figures and/or tables, and approved the final draft.
- Linlin Su performed the experiments, prepared figures and/or tables, authored or reviewed drafts of the paper, and approved the final draft.
- Xuetao Han analyzed the data, prepared figures and/or tables, and approved the final draft.
- Huandi Zhou analyzed the data, prepared figures and/or tables, and approved the final draft.

- Liubing Hou analyzed the data, prepared figures and/or tables, and approved the final draft.

## Data Availability

Raw measurements are available in the Supplemental Files.

## Supplemental Information

Supplemental information for this article can be found online at http://dx.doi.org/10.7717/peerj.12146#supplemental-information.

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
