# Peer review of "DNMT family induces down-regulation of NDRG1 via DNA methylation and clinicopathological significance in gastric cancer"

_PeerJ, doi:10.7717/peerj.12146_

## Round 0.1 · original submission · Major Revisions

Please fully address all the comments and concerns from all 3 reviewers.

Reviewer 1 ·

Basic reporting

This article is about the methylation genes in gastric cancer. This article is trying to elucidate the roles of these genes in gastric cancer.

Experimental design

The overall design of this study is quite general. The report of the methods needs improvement.

Validity of the findings

The validity of the results is not so reliable nor supported by their data. The findings and conclusions were not well supported.

Additional comments

This article is about the methylation genes in gastric cancer. This article is trying to elucidate the roles of these genes in gastric cancer. The overall design of this study is quite general. The report of the methods needs improvement. The validity of the results is not so reliable nor supported by their data. The findings and conclusions were not well supported.
Thus, I suggest rejecting this paper. One main concern are that the correlation coefficient is minimal. I also suggest the author add some gastric cancer samples to validate their findings.

1. Line 34, for the correlation, the author should report both p-value and correlation coefficient. The p=0.88 or p=0.57 for DNMT3B should not be regarded as significant.
2. Figure 2b, please specify the meaning of “**” .
3. In figure 3, the correlation coefficient is small to support the correlations. A p-value less than 0.05 is significant but does not guarantee correlation.
4. Line 120-121, please specify the type of correlation, e.g., Pearson correlation.
5. Figure 3e, delete the extra “?”
6. Figure 3e, the slope of the line seems inconsistent with the r-value.
7. There are also some typos and unannotated elements of figures.

·

Basic reporting

Chang et al investigate the correlation of NDRG1 with DNA methyltransferases in patients with gastric cancer. The authors utilize publicly available patient data and cell lines to support their findings.

Nevertheless, there are several parts that need significant improvement.

Experimental design

First of all, the authors in a biased approach, focus their efforts on the correlation of NDRG1 levels with the DNMT family, due to the importance in DNA methylation of CpG islands.

Given that the DNMTs are not solely responsible for the DNA methylation of CpG islands, the authors should also include their analysis in several components of PRC complex, expecially KDM2B.

Validity of the findings

The authors should include more publicly available datasets from GEO to improve the power of their study since the statistical significance of their findings are very weak.

More importantly, in the cell line data (Figure 2), the authors should use shRNA and inhibitors (decitabine) of DNMT1,3a,3b (and probably KDM2B) and overexpression vectors of the same proteins, in order to demonstrate the direct effect of DNMTs in the expression of NDRG1.

With their data they provide correlation, not causation.

Furthermore, in the same cells the authors should show the mRNA levels of NDRG1. To further prove the effect of DNMTs in NDRG1, the authors should use luciferase constructs of NDRG1 promoter and transfect them in shRNA against DNMT or DNMT inhibitors.

These data will support the claims of the authors.

Additional comments

Chang et al proposed the possible correlation of NDRG1 with DMNT DMA methyltransferase in patients with gastric cancer. The manuscript needs significant revision. I would be delighted the review the revised version of this interesting work. I would like to wish the authors good luck with their efforts.

·

Basic reporting

Overall, the reporting is scientific, clear and unambiguous. Sufficient background and references have been provided as well. However, here are some suggestions,
1. There are some grammatical and sentence construction errors that can be changed by getting it reviewed professionally. For example, line 39, 46, 117, 178 and so on.
2. The use of abbreviations are commendable and makes reading an article more pleasant. However, please introduce abbreviations in their first occurrence and following which only the abbreviations should be used in the rest of the article.
3. Only a couple of sentences are missing relevant references. For example, line 56, 174, 202.
4. Consider inserting a reference to the figure number when discussing it in the discussion for more concise understanding for the reader.

Experimental design

The experimental design is sound, relevant and meaningful.

Validity of the findings

The findings have been clearly stated along with underlying data have been provided. However, here are a few suggestions to consider for the discussion,
1. Consider addressing the claim in line 187 to address the challenges that can arise from comparison of in-vivo to in-vitro experimental assays. Consider expanding on probable future directions for the project as explained in brief in line 261.
2. Additionally, adding more relevant discussion on the role of DNA methylation can add value to discussion overall.

Additional comments

Overall, this article communicates the aim of the study, methodology and results very well to its readers.

---

## Round 0.2 · Minor Revisions

There are grammatical errors throughout the manuscript. Please correct ALL grammatical errors, and resubmit.

·

Basic reporting

Chang et al., provide a revised manuscript of their recent study on the regulation of NDRG1 by DNMT mediated DNA methylation in gastric cancer. The authors include several of the proposed experiments and fix issued noted. In my opinion, the revised manuscript is profound and strong. I would support the publication of this interesting concept.

Experimental design

No comments

Validity of the findings

No comments

Additional comments

Congratulation to the authors for their efforts.

---

## Round 0.3 · accepted · Accept

Concerns about typographical and grammatical errors have now been largely addressed.